# Optimizing TMS Coil Placement Approaches for Targeting the Dorsolateral Prefrontal Cortex in Depressed Adolescents: An Electric Field Modeling Study

**DOI:** 10.3390/biomedicines11082320

**Published:** 2023-08-21

**Authors:** Zhi-De Deng, Pei L. Robins, Moritz Dannhauer, Laura M. Haugen, John D. Port, Paul E. Croarkin

**Affiliations:** 1Computational Neurostimulation Research Program, Noninvasive Neuromodulation Unit, Experimental Therapeutics and Pathophysiology Branch, National Institute of Mental Health, Bethesda, MD 20892, USA; pei.robins@nih.gov (P.L.R.); moritz.dannhauer@nih.gov (M.D.); 2Department of Neurosurgery, Mayo Clinic, Rochester, MN 55905, USA; haugen.laura2@mayo.edu; 3Department of Radiology, Mayo Clinic, Rochester, MN 55905, USA; port.john@mayo.edu; 4Mayo Clinic Depression Center, Department of Psychiatry and Psychology, Mayo Clinic, Rochester, MN 55905, USA; croarkin.paul@mayo.edu

**Keywords:** adolescent, adult, major depressive disorder, treatment-resistant depression, treatment, transcranial magnetic stimulation, computational modeling, finite element analysis, dorsolateral prefrontal cortex, electromagnetic fields

## Abstract

High-frequency repetitive transcranial magnetic stimulation (rTMS) to the left dorsolateral prefrontal cortex (L-DLPFC) shows promise as a treatment for treatment-resistant depression in adolescents. Conventional rTMS coil placement strategies include the 5 cm, the Beam F3, and the magnetic resonance imaging (MRI) neuronavigation methods. The purpose of this study was to use electric field (E-field) models to compare the three targeting approaches to a computational E-field optimization coil placement method in depressed adolescents. Ten depressed adolescents (4 females, age: 15.9±1.1) participated in an open-label rTMS treatment study and were offered MRI-guided rTMS five times per week over 6–8 weeks. Head models were generated based on individual MRI images, and E-fields were simulated for the four targeting approaches. Results showed a significant difference in the induced E-fields at the L-DLPFC between the four targeting methods ($\chi^{2}=24.7$, p<0.001). Post hoc pairwise comparisons showed that there was a significant difference between any two of the targeting methods (Holm adjusted p<0.05), with the 5 cm rule producing the weakest E-field (46.0±17.4V/m), followed by the F3 method (87.4±35.4V/m), followed by MRI-guided (112.1±14.6V/m), and followed by the computational approach (130.1±18.1V/m). Variance analysis showed that there was a significant difference in sample variance between the groups ($K^{2}=8.0$, p<0.05), with F3 having the largest variance. Participants who completed the full course of treatment had median E-fields correlated with depression symptom improvement (r=−0.77, p<0.05). E-field models revealed limitations of scalp-based methods compared to MRI guidance, suggesting computational optimization could enhance dose delivery to the target.

## 1. Introduction

The Substance Abuse and Mental Health Services Administration (SAMHSA) reported that in 2020, 17% (or 4.1 million) of adolescents in the United States between the ages of 12 and 17 experienced a major depressive episode in the past year, among which 41.6% received treatment for depression in the past year [1]. Trends in suicide attempts and deaths by suicide have also been increasing among adolescents. For adolescents with treatment-resistant depression (TRD) and suicidal thoughts, a safe, tolerable, and promising treatment is repetitive transcranial magnetic stimulation (rTMS) [2,3,4,5,6]. The parameter space for dosing rTMS is vast and includes the: stimulation target, coil targeting strategy, frequency, intensity, train duration, interstimulus intervals, pulses per session, number of sessions, and brain state. Of these parameters, a major area of interest is determining optimal scalp coil placement for the left dorsolateral prefrontal cortex (L-DLPFC) [7]. Clinical studies on rTMS for adolescent depression have adopted coil placement approaches used in adults [2,8,9,10]. However, adult coil placement approaches may not yield optimal dosing and clinical outcomes in adolescents. For instance, there are structural differences between adults and adolescents in head anatomy, including head size and myelination development, which can differentially affect the spread of an induced electric field (E-field) in the brain. The rTMS-induced E-field can be modeled in individual subjects using computational methods [11]. These computational models can enhance our understanding of the effect of neurodevelopmental variability and have utility in the individualization of rTMS dosing and more suitable coil placement.

Approaches to TMS coil placement include: (1) the “5 cm rule”, (2) the Beam F3 method, (3) MRI guidance, and (4) computational approaches. Below, we discuss the advantages and limitations of each method. The 5 cm rule is the standard approach to placing a TMS coil over the L-DLPFC in adults and involves measuring the scalp 5cm along the parasagittal plane anterior to the activation hotspot in the motor cortex. While this method is easy to implement, it does not account for variations in head size, i.e., geodesic distances on scalp surface, between individuals. It has been shown that the 5 cm rule missed Brodmann area (BA) 9 in the L-DLPFC in more than two-thirds of adult subjects; the coil center often ended up over in more dorsal regions such as BA6 or 8 [12]. Further, while adults’ head sizes remain relatively fixed, adolescents’ head circumferences increase with age [13], resulting in a wider range of head sizes. Therefore, when applying the 5 cm rule in adolescents, there is an increased likelihood of missing the L-DLPFC target and hence producing more variability in the E-field dose and clinical outcomes. For example, in a blinded, randomized, sham-controlled clinical study on the effectiveness of high-frequency rTMS for TRD in adolescents, the 5 cm rule was used to target the L-DLPFC. The trial found no statistically significant difference between the active and sham groups in antidepressant efficacy [14].

The Beam F3 method is a popular approach that uses the International 10–20 electroencephalogram (EEG) system for positioning of TMS; in particular, the F3 site corresponds to parts of BA8, 9, and 46 within the L-DLPFC [15,16]. Beam and colleagues developed an efficient way to locate the F3 position from a series of scalp measurements [17]. Compared to the 5 cm rule, the Beam F3 method better scales with head size since the EEG electrode-placement system is based on measurements of head circumference and nasion–inion and tragus–tragus distances.

In addition to the scalp-based methods, MRI neuronavigation is an alternative to localize the L-DLPFC based on individual brain images. In our prior rTMS study for depressed adolescents, the 5 cm rule and the Beam F3 method yielded different locations that were on average 3.9cm and 2.5cm away from the MRI-derived L-DLPFC scalp target, respectively [9]. In adults, rTMS delivered with the MRI-based approach resulted in superior outcomes compared to the 5 cm rule; however, MRI acquisition is costly and therefore less practical for clinical TMS [18]. Mir-Moghtadaei and colleagues suggested that the Beam F3 method may provide a reasonable approximation of MRI-guided neuronavigation in a majority of adults [19].

The above-mentioned coil placement methods do not consider the induced E-field. Computation modeling can help to determine the TMS coil placement that maximizes the delivered E-field in the targeted brain region. A previous computational study showed that, compared to simply placing the coil above the center of mass of the target, the optimal scalp coil placement can be more than 10mm away, leading to an E-field strength approximately 6% higher [20]. Our Targeting and Analysis Pipeline (TAP) was subsequently developed and further demonstrated how a voxel-based ROI can be efficiently integrated with TMS coil placement optimization [21].

The goal of this study is threefold. First, we aim to quantify and compare the induced E-field by three targeting approaches (5 cm rule, F3 method, and MRI-based method) in a group of depressed adolescent participants who have previously undergone an acute course of high-frequency rTMS delivered to the L-DLPFC. Secondly, we examine the inter-individual variability of the delivered E-field and assess its relationship with the treatment outcome. Lastly, we explore the utility of a computational approach [21] for optimizing individual coil placement to maximize the induced E-field in the L-DLPFC. This study serves as an important initial step in a larger effort to enhance rTMS protocols for the treatment of adolescent depression.

## 2. Methods

This study is a computational modeling investigation based on data collected from a previously reported open-label rTMS treatment study [8,9]. Here, we provide a concise overview of the essential trial procedures encompassing participants’ demographics, motor hotspot and resting motor threshold determination, neuroimaging parameters, and treatment intervention. Subsequently, we detail the E-field modeling methods, covering head model generation, E-field computation, DLPFC mask definition, coil placement and optimization, estimation of device output intensity, and postprocessing techniques.

### 2.1. Participants

This open-label rTMS study was conducted under an Investigational Device Exemption (#G110091) from the United States Food and Drug Administration and was approved by the Mayo Clinic Institutional Review Board (ClinicalTrials.gov identifier: NCT01502033). Ten Caucasian participants (four females) with treatment-refractory major depressive disorder (MDD) and between the ages of 13.9 and 17.4 years (mean±standarddeviation=15.9±1.1) participated in an open-label rTMS treatment study after providing informed consent and assent. Each patient met the Diagnostic and Statistical Manual of Mental Disorder, Fourth Edition, Text Revision (DSM-IV-TR) criteria for a major depressive episode based on a semi-structured diagnostic interview: the Schedule for Affective Disorders and Schizophrenia for School-Age Children-Present and Lifetime version (K-SADS-PL) [22]. Participants had moderate-to-severe symptom severity ratings as evidenced by a baseline Children’s Depression Rating Scale-Revised (CDRS-R) [23]. A total CDRS-R score of 40 or greater at baseline was required for inclusion criteria. Further, all patients had to have at least one prior failed antidepressant medication trial as defined by the Antidepressant Treatment History Form [24].

### 2.2. Motor Hotspot and Resting Motor Threshold Determination

Motor hotspot and motor threshold were determined from visual observation of movement in the *abductor pollicis brevis* (APB) muscle of the right hand during TMS over the contralateral motor cortex using the NeuroStar coil (Neuronetics, Inc., Malvern, PA, USA). MT was measured in units of Standard Motor Threshold (SMT) and was defined as the lowest stimulator setting at which greater than 5 out of 10 stimuli resulted in any observable movement of the right thumb.

### 2.3. Imaging

Patients were custom-fitted with a swim cap on which the APB motor hotspot, 5 cm site, F3 site, mid-frontal, left mastoid, right mastoid, right parietal, and right frontal were marked with fiducial markers as described previously [9]. T1-weighted structural MRI data were acquired on a GE 3-T DV750 scanner equipped with an eight-channel head coil (true-axial fast 3D-SPGR sequence: repetitiontime(TR)=12.6ms, echotime(TE)=5.6ms, flipangle=15degrees, $\mathrm{voxel}\mathrm{dimensions}=0.49\times 0.49\times 1.5\;{\mathrm{mm}}^{3}$, $\mathrm{field}\;\mathrm{of}\;\mathrm{view}=250\times 250\;{\mathrm{mm}}^{2}$, slice=1.5mm, and matrix=512×250pixels).

### 2.4. Treatment Intervention

The treatment target was derived from the T1-weighted anatomical images using the Medtronic StealthStation™ navigation system (Medtronic Navigation, Inc., Louisville, CO, USA) by creating real-time surgical navigation on patients’ radiological images (see Appendix A for coordinate transformation). The DLPFC brain target (DBT) was defined as a $20\times 20\times 20\;{\mathrm{mm}}^{3}$ voxel in the L-DLPFC according to the following anatomical guidelines: (1) the “inferior plane” of the corpus callosum was identified as a line that abutted the inferior margins of the rostrum and splenium of the corpus callosum; (2) a 20 mm thick coronal–oblique localizer slice was acquired perpendicular to the inferior plane, such that the center of the localizer slice was placed 10 mm anterior to the genu of the corpus callosum and the posterior edge of the slice abutted the anterior margin of the rostrum of the corpus callosum; (3) the deepest portion of the superior frontal sulcus was identified to be the DBT. The averaged center of the DBT voxels was projected through the shortest straight path to the scalp, yielding the coordinates for the DLPFC scalp target (DST) [9]. Over the DST, the standard high-frequency rTMS protocol (10 Hz freqeuncy, train duration of 4 s, intertrain interval of 26 s, and 3000 pulses per session) was delivered using the NeuroStar Therapy System at an intensity of 120% SMT 5 days per week over 6–8 weeks up to a total of 30 sessions.

### 2.5. E-Field Modeling

#### 2.5.1. E-Field Computation and Head Model Generation

Due to the low-frequency nature of TMS pulses, TMS E-field simulations are modeled in the quasi-static regime of electromagnetics [25]. In this regime, conduction currents do not result in inductive coupling. The E-field solver, implemented in the SimNIBS software [26], calculates the total E-field by considering the coil-emitted magnetic field pulse and the resulting charge build-up. The charge build-up occurs in regions where resistive tissue properties change, and it is represented by assigning electrical conductivities to each computational element, which corresponds to a small piece of the tissue being modeled. To segment different head tissues, the SimNIBS software utilizes T1-weighted MRI data sets in the chosen mri2mesh pipeline. Before the pipeline begins, fiducial markers are manually removed from the T1-weighted anatomical images using ImageJ [27]. In the mri2mesh process, the surfaces of the tissues are smoothed and filled with tetrahedral elements to create the computational head mesh. Each tetrahedral element is assigned an isotropic conductivity based on the corresponding tissue type: skin, skull, cerebrospinal fluid, gray matter, white matter, and eyes have conductivities of 0.465S/m, 0.010S/m, 1.654S/m, 0.275S/m, 0.126S/m, and 0.5S/m, respectively.

The correctness, performance, and numerical accuracy of the finite element method (FEM) utilized in this work have been validated previously by comparison to an analytical exact solution for a spherical head model [28]. The validation demonstrated the reliability and accuracy of the FEM approach. Moreover, the validity of the TMS E-field simulations conducted using the same modeling pipeline and software has been established through the observation of high correlations between the strength of the E-field and TMS-evoked physiological measurements of the motor cortex [29].

#### 2.5.2. Structural Mask Definition

The L-DLPFC mask was binarized (MRIcroGL, https://www.nitrc.org/projects/mricrogl/ (accessed on 14 March 2022)) using the skull-stripped MNI-152 template and defined by the following procedures: (1) measure the distance between the most anterior point of the frontal pole and the most anterior ipsilateral temporal pole ($d_{\mathrm{FP}–\mathrm{TP}}$); (2) measure the distance from the tip of the temporal pole anteriorly 20% $d_{\mathrm{FP}–\mathrm{TP}}$ to mark the posterior vertical boundary of L-DLPFC; (3) measure the distance from the tip of the frontal pole posteriorly 40% $d_{\mathrm{FP}–\mathrm{TP}}$ to mark the anterior vertical boundary of L-DLPFC; (4) measure the distance from the most inferior part of the temporal lobe to the most superior part of the brain ($d_{\mathrm{BB}–\mathrm{TB}}$); and (5) in the coronal slice, measure the distance from 50% $d_{\mathrm{BB}–\mathrm{TB}}$ superiorly to mark the inferior boundary of the superior frontal sulcus [30,31]. The L-DLPFC mask in the MNI space was transformed into the subject’s space using the FLIRT (FMRIB’s Linear Image Registration Tool) function in FSL with 12 degrees of freedom [32].

#### 2.5.3. Coil Placement and Optimization

The Neuronetics coil was positioned at the center of the DST treatment target and oriented 45° toward the hemispheric midline. In addition to simulating the E-field at the DST treatment target, we also conducted simulations with the coil placed over the APB motor hotspot and hypothetical treatment targets (5 cm and F3 locations) identified from the MRI. In the simulations, we used the default value of 4 mm for hair thickness, which represents the distance between the scalp and the coil.

To determine the coil placement within the L-DLPFC that would maximize the delivered E-field, we employed a computational optimization approach. Since there was no dipole coil file available for the Neuronetics coil, we utilized a direct solver implemented in SimNIBS. The L-DLPFC mask was first registered to the subject using the TAP software [21]. This process allowed us to determine the center voxel coordinate and size of the ROI, represented as a sphere. On average, the individual ROIs had a spherical radius of 11.4±0.48mm. The E-field optimization involved a discrete search, where different coil centers (on a 1 mm grid) and orientations (with 4° increments, resulting in over 50,000 independent simulations) were evaluated based on the maximum averaged E-field magnitude within a 20-mm radius around the scalp-projected point of the L-DLPFC center. Although this direct approach for coil placement optimization (using the PARDISO solver) was significantly slower (taking 11 s per TMS simulation on our high-performance computing cluster, with a total computation time of approximately 1 week per subject) compared to SimNIBS-implemented alternatives (such as ADM, which takes approximately 1 h of total computation time, [20]), it provided higher numerical accuracy by performing over 50,000 TMS simulations per subject.

For all coil placements, the E-field was simulated using the rate of change of the coil current, dI/dt, corresponding to the individual treatment dose in SMT units.

#### 2.5.4. Estimation of dI/dt for the Standard Motor Threshold

The output of the NeuroStar System in terms of the rate of change of coil current, dI/dt, was proprietary. The stimulator output level units are Standard Motor Threshold (SMT) units. According to the device technical data sheet, 1 SMT is the output setting that corresponds to an induced E-field of 135 V/m at a point located 2 cm along the central axis of the treatment coil from the surface of the scalp into the patient’s cortex [33]. This corresponds to the average motor threshold level observed in a large patient population [33]. With this information, we converted the SMT unit to dI/dt using a spherical head model, as was done in previous work [34]. The head was modeled as a homogeneous sphere with a radius of 8.5 cm and isotropic conductivity of 0.33 S/m.

The Neuronetics coil consisted of a figure-eight winding with a C-shaped ferromagnetic core [34,35] (Figure 1A). The spatial E-field distribution was computed for an arbitrary coil current $I_{0}=1\;A$ and frequency $\omega_{0}=2\pi \times 5\;\mathrm{kHz}$ using the MagNet Time Harmonic solver, yielding an E-field value at 2 cm depth of $E_{2\;\mathrm{cm}}^{\prime }=0.066\;V/m$ (Figure 1B). The dI/dt was subsequently calculated for the desired field strength of 135 V/m at 2 cm depth:$$\begin{array}{cc} E_{2\;\mathrm{cm}} & =\frac{E_{2\;\mathrm{cm}}^{\prime }}{\omega_{0}I_{0}}\;\cdot \;\frac{dI}{dt},\\ 135\;V/m & =\frac{0.066\;V/m}{2\pi \times 5\;\mathrm{kHz}\times 1\;A}\;\cdot \;\frac{dI}{dt},\\ \frac{dI}{dt} & =64.67\;A/\mu s. \end{array}$$

Using this dI/dt value, the mean maximum induced E-field at the motor cortex of the adolescent subjects at the individual motor threshold is 141.3±14.7V/m, which is comparable to the reference SMT field strength of 135 V/m reported for adults (t=1.30, p=0.23).

#### 2.5.5. E-Field Postprocessing

The SimNIBS msh2nii command-line tool was used to interpolate simulated E-field magnitude values for gray matter MRI voxels within the L-DLPFC mask. The median E-field strength was extracted from the L-DLPFC gray matter voxel mask. The median E-field values from the four targeting methods were compared using the Friedman test followed by Post hoc pairwise Wilcoxon tests with Holm adjustment. Finally, a recent report suggested that the normal component of the TMS-induced E-field is correlated with depressive symptom relief in treatment-resistant depression in adults [36]. Thus, we explored the relationships between the median magnitude of the E-field and its normal component with the change in CDRS for participants who completed a full course of treatment. The E-field normal component was calculated by mapping the vectorized E-field and mask mesh files to the cortical surface (Freesurfer’s FsAverage surface) with the msh2cortex command-line tool, which utilizes the superconvergent path recovery method [37].

## 3. Results

### 3.1. Dose–Response Relationship between E-Field and Clinical Outcome

Of the ten participants enrolled in the study, one had an unusually high motor threshold (1.54SMT). This high threshold resulted in high treatment stimulation intensity (1.85SMT) and high median L-DLPFC E-field (214.0 V/m). The high stimulation intensity contributed to scalp discomfort that the participant was unable to tolerate; the participant dropped out of the study after the first rTMS session. The subsequent E-field analysis was performed with the data from the remaining nine participants. In addition to the participant who dropped out, two other participants did not complete the full course of rTMS treatments due to worsening depression and anxiety; their data were removed from the correlational analysis with E-field and clinical outcomes.

Figure 2 explores the relationship between the L-DLPFC E-field magnitudes and the change in CDRS scores in participants who received a full treatment course. The median E-field magnitude was linearly correlated with the change in CDRS (r=−0.77, p<0.05), while the normal component of the E-field did not show such a relationship (p>0.05).

### 3.2. Comparison between Alternative Coil Placements

Figure 3 shows the individual TMS coil placements and corresponding E-field distributions on the cortical surface.

Figure 4A shows the TMS coil placements mapped to the MNI space. The scalp area spanned by the 5 cm placements were mostly outside the bounds of the DLPFC ROI; the F3 placements were partially overlapping with the DLPFC; the DST treatment targets were mostly within the DLPFC; and the computationally derived placements were all directly over the DLPFC. Figure 4B shows the MNI locations of the induced E-field maxima. Note that the individual E-field maxima may not be directly under the center of the coil due to the effects of local gyrus geometry. The distances between the E-field maxima and the DLPFC centroid are: 31.9 mm for the 5 cm placements, 18.9 mm for the F3 placements, 8.2 mm for the DST placements, and 4.1 simm for the computationally derived placements.

There was a significant difference in the induced E-field at the L-DLPFC between the four targeting methods ($\chi^{2}=24.7$, p<0.001). Post hoc pairwise comparisons showed that there was a significant difference between any two of the targeting methods (Holm adjusted p<0.05), with the 5 cm rule producing the weakest E-field, followed by the F3 method, followed by the MRI-guided DST method, and followed by the computational method. The Bartlett test of homogeneity of variances showed that there was a significant difference in sample variance between the groups ($K^{2}=8.0$, p<0.05), with the F3 method having the largest variance in E-field strengths (Figure 5).

In summary, the modeling results indicate marked variability in the induced E-fields across individuals receiving rTMS to the L-DLPFC. There appears to be a linear correlation between the median E-field magnitude and the change in depression symptom scores, suggesting that the intensity of the E-field may have a significant impact on clinical improvement. Additionally, we compared alternative coil placements using four targeting methods and observed significant differences in the induced E-field at the L-DLPFC among these methods. The computational approach yielded the highest E-field strength, while the 5 cm rule produced the weakest E-fields. The results highlight the importance of accurate coil placement for optimizing the therapeutic effects of rTMS in treating depression in adolescents.

## 4. Discussion

In this study, we investigated the dose–response relationship between the E-field magnitude and clinical outcomes in the L-DLPFC for depressed adolescents who participated in an open-label rTMS treatment study. We identified a linear correlation between the median E-field magnitude and the change in CDRS scores, highlighting the importance of accurately targeting the L-DLPFC to maximize therapeutic effects. Additionally, we compared several TMS targeting strategies, including the 5 cm rule, Beam F3, MRI-guided DST, and a computationally derived method. Our findings indicated that the 5 cm rule consistently underdoses the L-DLPFC, potentially explaining negative results in recent adolescent rTMS trials. The Beam F3 method demonstrated improved accuracy, but its precision varied across individuals. These insights laid the foundation for the development of our new targeting protocol.

Analogous to Global Positioning System (GPS) navigation, there are two key components to TMS targeting: the first is to determine the location of the destination address (target identification); the second is to chart the path to the destination accurately and precisely (coil placement). The search for the optimal target for rTMS depression treatment has generated much recent research interest. The DLPFC is the standard targeting site used in rTMS studies for both adults and adolescents. The conventional rationale to target the DLPFC is based on its hemispheric asymmetry and imbalance of activities in depression, in which the left DLPFC is hypoactive and the right DLPFC is hyperactive [38,39]. To balance both hemispheres of the DLPFC, high-frequency rTMS (10–20 Hz) is used to increase neuronal excitability of the L-DLPFC, while low-frequency rTMS (1 Hz) is used to induce neuronal inhibition of the R-DLPFC [40]. Other approaches to identify a target within the DLPFC use functional neuroimaging: for example, using resting-state fMRI to determine the location of the maximum anticorrelation between the L-DLPFC and the subgenual anterior cingulate cortex (sACC) [41,42], or task-based fMRI to determine the location of the peak activation in engagement with a goal-priming task [43]. Another approach attempts to engage the DLPFC node in the frontal–vagal pathway that overlaps with functional nodes of the depression network [44]. The resultant Neuro–Cardiac-Guided TMS (NCG-TMS) technique uses TMS-induced heart rate deceleration as a marker for target engagement and stimulation site determination [45]. Finally, although the DLPFC has been the most popular targeting site, recent studies have attempted to target other regions that are involved in emotion regulation [46], such as the dorsomedial prefrontal cortex [47,48,49] and the right orbitofrontal cortex [50].

Once the treatment target has been identified, the other aspect of TMS targeting is to choose a navigation strategy that will best locate the stimulation site. The clinical standard for targeting the L-DLPFC has been the 5 cm rule. In this work, we showed that among targeting strategies, the 5 cm rule produces the lowest E-field in the L-DLPFC. In view of the dose–response function between E-field strength and antidepressant outcome (Figure 2), the consistent underdosing of the E-field by the 5 cm rule could lead to a suboptimal therapeutic effect, which could partly explain the negative results in the recent adolescent rTMS trial [14]. There have been proposed variations to the 5 cm rule to improve its accuracy, such as the 5.5 cm rule [51] and the “5cm+1cm” (6 cm) rule [52]. Johnson and his colleagues compared the targeting variations using the 5 cm rule and the 6 cm rule [52]. They found that both targeting strategies produced similar intra-variability across adult subjects, in which the 6 cm rule would shift the variability range more anterior. By using the 6 cm rule, they proposed that this variation could offer more therapeutic effects. However, the authors did not compare the clinical efficacy between these targeting variations [52]. An additional limitation of the 5 cm rule concerns the motor hotspot and threshold determination technique, which was done with observed movement of the target muscle method (OM-MT) without electromyography (EMG). Prior work suggests that the OM-MT method yields significantly higher MTs compared to the thresholding method based on EMG [53]. The OM-MT method can produce a suboptimal motor hotspot to which the 5 cm target is anchored to and elevated risk due to overestimation of the MT to which the treatment stimulation intensity is scaled to.

The Beam F3 method is another navigation strategy to target the L-DLPFC. In our E-field modeling, we showed that compared to the 5 cm rule, the Beam F3 method can achieve more accurate targeting, albeit not necessarily more precise. That is, on average, the Beam F3 method produces higher E-fields in L-DLPFC, but in our sample, it showed more variability across individuals. A recently published report showed that in adults, when comparing coil distance from BA46, the Beam F3 method produced the largest variance from the target compared to the 5 cm rule and the MRI-guided approach [54]. There have been proposed modifications to the Beam F3 method. For example, one study modified the method to better estimate the optimized anti-subgenual TMS target [55] using the MNI anatomical template to determine the distances (measurements that would be required) from anatomical landmarks to the target average coordinate that showed the greatest DLPFC–sACC anticorrelation. This variation showed that the newer anterior L-DLPFC estimate was 21.5±1.4mm more inferior–posterior to F3, while the posterior L-DLPFC was 37.0±0.6mm more posterior to F3.

MRI neuronavigation is another targeting strategy with different adaptations. Structural MRI-guided methods for localizing the DLPFC can be based on targeting specific Brodmann area sites or using other anatomical definitions. The DLPFC comprises two different cyto-architectural sub-regions: BA9 and 46. A preliminary study randomly assigned participants to two treatment groups to receive either rTMS over BA9 or over BA46: stimulation of both Brodmann areas led to similar antidepressant responses [56]. Other studies have targeted a site in the junction of BA9 and 46 using coordinates in a standard atlas space such as the Talairach atlas [57]. Our study defined the L-DLPFC as a point projection of a voxel in the brain based on a series of anatomical guidelines [8,9] as described in the Methods section. Functional MRI-guided targets can be derived from group-averaged functional maps [58,59,60,61] or from individualized connectivity [62,63]. The optimal treatment target can vary across individuals; another factor that contributes to variability is inter-individual differences in head anatomy, which can influence the spatial distribution of the induced E-field [64]. In this work, we showed that given a target of interest, computational optimization can be used maximize the E-field delivery to the target. More-sophisticated algorithms have been proposed to combine individual functional connectivity patterns and E-field optimization to determine coil placements that would not only maximize local stimulation but also account for downstream effects of TMS, i.e., to maximize brain network engagement [65,66,67,68]. Systematic clinical trials are needed to prospectively compare the antidepressant efficacy of these strategies in adults and adolescents.

There are a number of limitations to consider for the interpretation of this work. First, this is a pilot study with small a sample size with participants having variable treatment response. This is exacerbated by drop-outs–several participants were unable to complete the full treatment course. Although data from the remaining participants did suggest a dose–response relationship (Figure 2), this is to be interpreted with caution. The second limitation concerns the head model tissue properties used in the E-field simulations, particularly in developing adolescents. During adolescence, the brain undergoes significant changes in terms of both structure and function. One of the key changes that occurs during this time is the continuation of myelination that helps to refine and optimize the connectivity between different brain regions. Studies have shown that myelination in adolescents is lower—particularly in the frontal lobes—compared to adults [69]. Differences in the degree of myelination may, in part, explain age differences in motor thresholds, which are higher in younger individuals [70]. When modeling E-fields in the head and brain, it is common to assign standard isotropic conductivity values for different tissues based on values that have been reported in the literature; this tissue-property assignment does not factor in the effects of age. Finally, our study does not address the effects of broad stimulation of the DLPFC with nonfocal TMS coils.

## 5. Conclusions

Proper placement of the TMS coil is necessary to ensure that the desired brain region is adequately stimulated while minimizing the risk of stimulating unintended areas. Understanding the factors that influence delivered E-field dose in the brain can help to improve the effectiveness of rTMS as a treatment. In this computational study, we modeled a group of adolescents receiving rTMS targeted to the L-DLPFC. We made within-subject comparisons of three targeting strategies: the 5 cm rule, the Beam F3 method, and MRI-guided targeting. These models showed various shortcomings of the scalp-based targeting methods: the 5 cm rule underdosed E-fields to the L-DLPFC, and the Beam F3 method exhibited higher inter-subject variability. Both of these scalp-based targeting methods produced significantly lower E-fields compared to MRI-guided targeting. In the study, for participants who received a full course of MRI-guided rTMS, the data suggested a dose–response relationship between the E-field strength in the L-DLPFC and clinical improvement. We further showed that computational optimization can be used to maximize the E-field dose. This motivates the use of computational techniques to further optimize E-field delivery in future clinical trials.

## Figures and Tables

**Figure 1 biomedicines-11-02320-f001:**
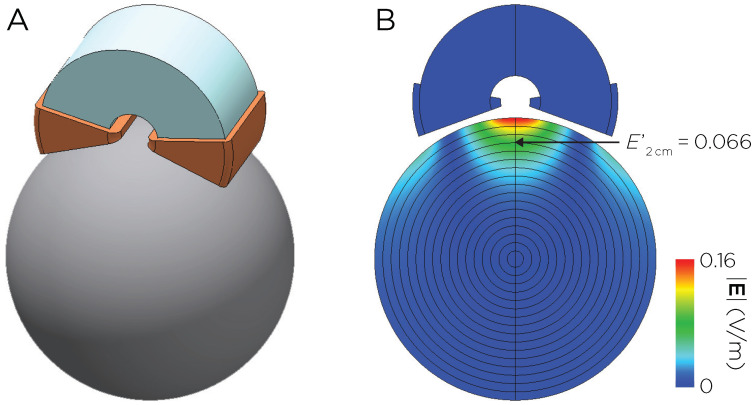
(**A**) Model of the Neuronetics coil with C-shaped ferromagnetic core. (**B**) Induced E-field distribution in a spherical head model with unit coil current $I_{0}=1\;A$ and frequency $\omega_{0}=2\pi \times 5\;\mathrm{kHz}$.

**Figure 2 biomedicines-11-02320-f002:**
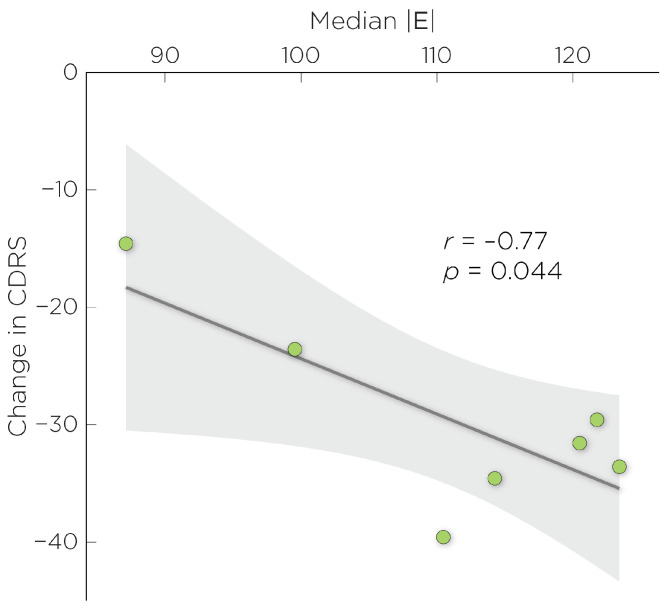
Correlation between the change in CDRS with median E-field magnitude in the L-DLPFC.

**Figure 3 biomedicines-11-02320-f003:**
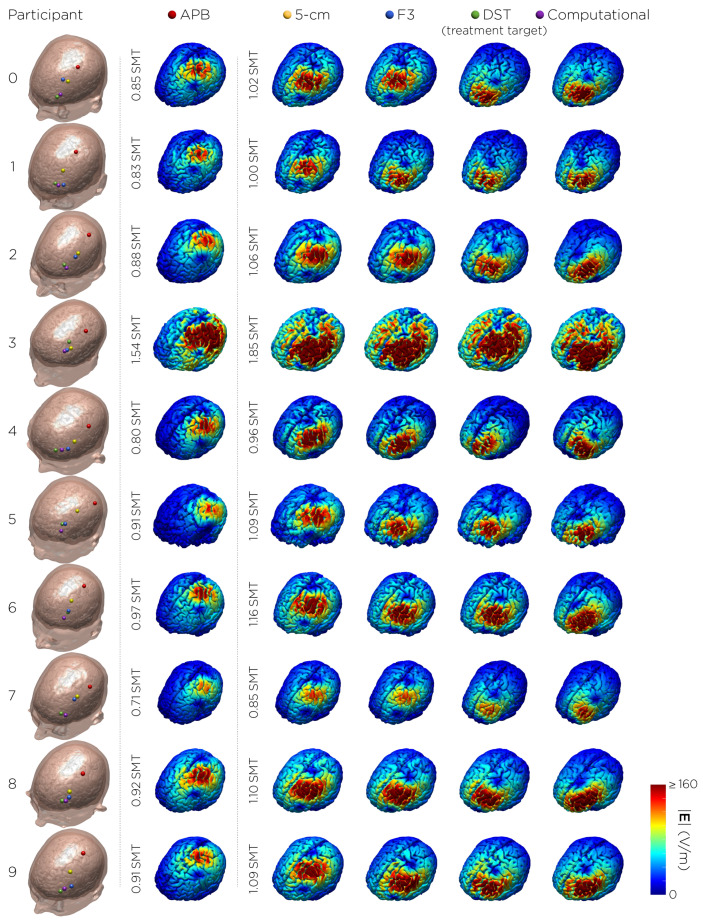
TMS coil placements and corresponding E-field distributions on the cortical surface. The E-fields for APB motor hotspot (red dot) stimulation were simulated with input intensities corresponding to individual motor threshold (SMT) units. The E-fields for the treatment targets (5 cm rule (yellow), Beam F3 (blue), MRI-derived DLPFC Scalp Target (green), and computationally derived coil placement (purple)) were simulated at 120% motor threshold. One of the participants (#3) dropped out of the study after the first session due to unusually high motor threshold and treatment stimulation intensity.

**Figure 4 biomedicines-11-02320-f004:**
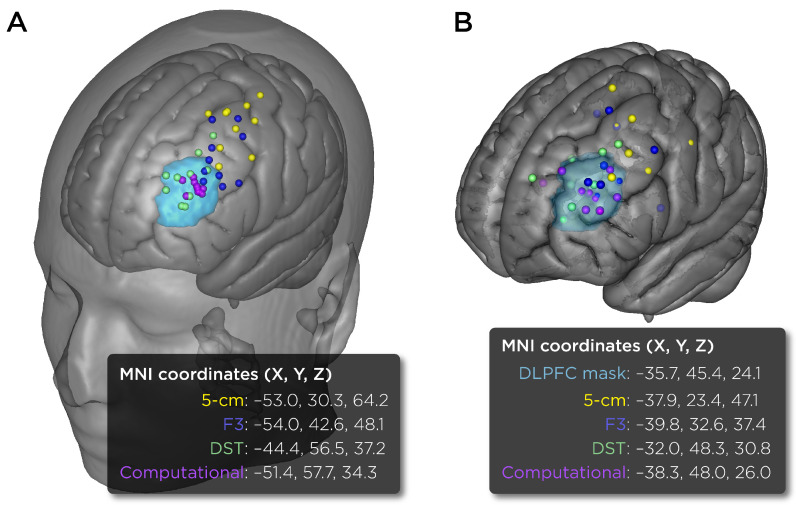
TMS coil placements mapped to MNI space via nonlinear transformation. (**A**) Scalp coordinates for the coil placements (5 cm rule (yellow), Beam F3 (blue), MRI-derived DLPFC Scalp Target (DST treatment target, green), and computationally derived coil placement (purple)). The average MNI coordinates for the group are provided in the text box. (**B**) Brain coordinates for the locations of E-field maxima. The DLPFC mask is highlighted in turquoise color. Visualizations were created using SCIRun 4.7 (R458390).

**Figure 5 biomedicines-11-02320-f005:**
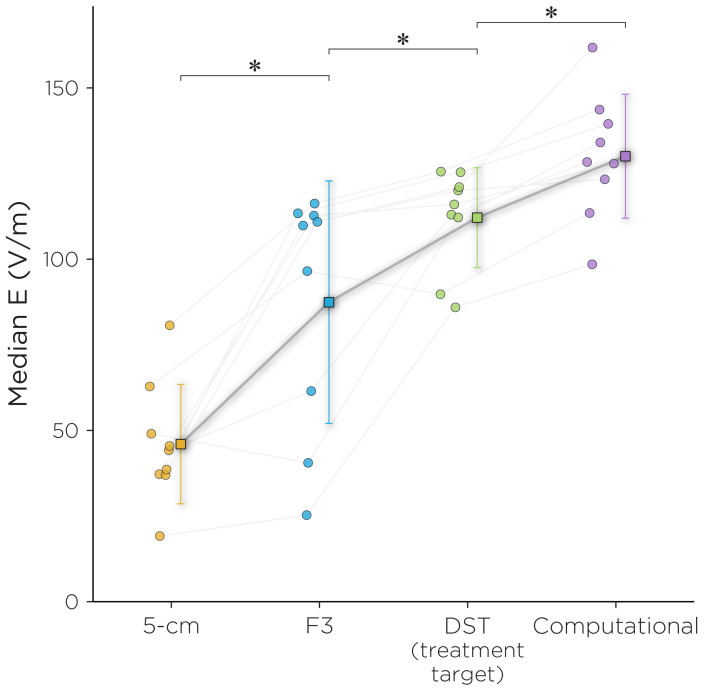
Median E-fields for the four targeting strategies. Post hoc pairwise comparisons showed that there was a significant difference between any two of the targeting methods (∗ indicates Holm adjusted p<0.05). The inset shows L-DLPFC voxel mask on which the median E-field was extracted.

## Data Availability

The data presented in this study are available on request from the corresponding author.

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
