# Peer review of "Optimizing TMS Coil Placement Approaches for Targeting the Dorsolateral Prefrontal Cortex in Depressed Adolescents: An Electric Field Modeling Study"

_biomedicines, 2023, doi:10.3390/biomedicines11082320_

Round 1
Reviewer 1 Report
I think this article is very important to optimize rTMS effects on depression in future. Although this study was an open study with small sample size, I think this manuscript would be suitable for publication.
Author Response
Thank you.
Reviewer 2 Report
The authors presented the proofs of promising treatment option model for depression in adolescents by means of an electric field (E-field) to compare the three targeting approaches to a computational E-field optimization coil placement method, instead of high-frequency repetitive transcranial magnetic stimulation (rTMS) delivered to the left dorsolateral prefrontal cortex (L-DLPFC). The latter is utilized with conventional coil placement strategies for rTMS in adults include the 5-cm rule, the Beam F3 method, and the magnetic resonance imaging (MRI) neuronavigation method.
Although only ten participants were tested, the results are reliable. The application of MRI-guided rTMS 5 times per week over 6-8 weeks seems numerous enough to ascertain the effects, so the study design is appropriate.
The authors carefully concluded, that compared to MRI guidance, the E-field models revealed inadequacies of scalp-based targeting methods, and computational optimization might further enhance E-field dose delivery to the treatment target.
No deficiencies but suggestions for the manuscript minor improvements can be provided:
I think that the data attached in the supplementary material file should be incorporated into the main text in the Results section.
Abstract: Adolescent; in keywords (line 23 ) is not necessary. I would be better to introduce instead the first keyword „depression treatment”.
In the Introduction, after a broad explanation of the study's aims., one short aim sentence after line 87 would clear the authors' intention for the practitioners interested in the therapy principles.
The M&M, Results and Discussion sections are clearly presented.
Limitations should include a factor of inter-individual variation.
Selection and citation of Refs. is perfect.
Reviewer 3 Report
Summary/Contribution: This study compares coil placement options for high-frequency repeated transcranial magnetic stimulation (rTMS) in depressed teenagers using E-field models. The study examines the 5-cm rule, Beam F3, MRI neuronavigation, and computational E-field optimization coil placement methods. The computational strategy produced the strongest E-field at the left dorsolateral prefrontal cortex (L-DLPFC) of the four targeting strategies. The L-DLPFC E-field strength is also connected with depression severity in those who completed treatment. MRI-guidance may be better than scalp-based targeting, and computational optimization may improve E-field dosage delivery to the treatment target.
Comments/Suggestions:
2. The introduction provides a good overview of the background and motivation for the study. However, it would be helpful to provide more context on the prevalence and impact of treatment-resistant depression in adolescents. This could help frame the significance of the study and why optimizing rTMS dosing for this population is important.
3. The section could benefit from clearer organization and structure. For instance, the authors could introduce the different coil placement strategies earlier in the section and then discuss the advantages and limitations of each method in separate subsections.
4. It would be helpful to provide more details on the methods used to generate the computational models and optimize coil placement. This could help readers understand the technical aspects of the study and the potential implications of the findings.
5. The authors could consider providing more information on the sample size and demographic characteristics of the participants.
7. More information, such as the number of trials and the criteria used to define the threshold, would be helpful in determining the motor hotspot and the resting motor threshold.
8. More information, including the logic behind target selection and details about the swim cap and fiducial markers used to designate the various brain areas, would be beneficial.
9. More context about the selection of individual rTMS treatment parameters, such as stimulation frequency and intensity, would be helpful.
10. More detail on the assumptions and restrictions of the E-field modeling approach utilized in the study, as well as how these may have affected the results, would be beneficial.
11. The authors may discuss the use of formal methods for checking the correctness of the proposed system. 12. For this purpose, the authors may include the following references:
a. https://link.springer.com/chapter/10.1007/978-3-030-51517-1_33
b. https://link.springer.com/chapter/10.1007/978-3-319-12214-4_15
May be improved
Author Response
Please see the attachment for a point-by-point response.

Round 2
Reviewer 3 Report
The authors considered my comments and suggestions.
Author Response
Thank you.